# PeerJ

# The PGI enzyme system and fitness response to temperature as a measure of environmental tolerance in an invasive species

Marie-Caroline Lefort[1], Samuel Brown[1], Stéphane Boyer[1,2], Susan Worner[1] and Karen Armstrong[1]

[1] Bio-Protection Research Centre, Lincoln University, Lincoln, Christchurch, New Zealand
[2] Department of Ecology, Faculty of Agricultural and Life Sciences, Lincoln University, Lincoln, New Zealand

## ABSTRACT

In the field of invasion ecology, the determination of a species' environmental tolerance, is a key parameter in the prediction of its potential distribution, particularly in the context of global warming. In poikilothermic species such as insects, temperature is often considered the most important abiotic factor that affects numerous life-history and fitness traits through its effect on metabolic rate. Therefore the response of an insect to challenging temperatures may provide key information as to its climatic and therefore spatial distribution. Variation in the phosphoglucose-6-isomerase (PGI) metabolic enzyme-system has been proposed in some insects to underlie their relative fitness, and is recognised as a key enzyme in their thermal adaptation. However, in this context it has not been considered as a potential mechanism contributing to a species invasive cability. The present study aimed to compare the thermal tolerance of an invasive scarabaeid beetle, *Costelytra zealandica* (White) with that of the closely related, and in part sympatrically occurring, congeneric non-invasive species *C. brunneum* (Broun), and to consider whether any correlation with particular PGI genotypes was apparent. Third instar larvae of each species were exposed to one of three different temperatures (10, 15 and 20 °C) over six weeks and their fitness (survival and growth rate) measured and PGI phenotyping performed via cellulose acetate electrophoresis. No consistent relationship between PGI genotypes and fitness was detected, suggesting that PGI may not be contributing to the invasion success and pest status of *C. zealandica*.

Corresponding author
Marie-Caroline Lefort, Marie-Caroline.Lefort@lincolnuni.ac.nz

## INTRODUCTION

Understanding environmental tolerance is a key factor in predicting a species' potential geographic and ecological range. This in turn is important for the successful management of invasive species. In insect species thermal tolerance is especially important, with extreme temperatures known to affect their development and influence their population dynamics (*Wallner, 1987*; *Clarke, 2003*; *Sinclair, Williams & Terblanche, 2012*). Direct effects on

metabolic rate impact on a number of life-history and fitness traits (*Clarke, 2003*; *Karl, Schmitt & Fischer, 2008*), among which survival and growth (*Folguera et al., 2010*) are the most important. For instance, *McMillan et al. (2005)* reported a significant increase in larval mortality of the leaf beetle *Chrysomela aeneicollis* Schaeffer in the coldest of three river drainages tested, which was exposed to subzero night-time air temperature. Similarly, *Kallioniemi & Hanski (2011)* reported low survival rates with larvae of the Glanville fritillary butterfly *Melitaea cinxia* (Linnaeus) subjected to low temperature stress. High temperatures can be equally detrimental as demonstrated by *Papanikolaou et al. (2013)*, where development of the immature stages of the 14-spotted ladybird beetle *Propylea quatuordecimpunctata* (Linnaeus) was greatly impaired and high mortality rates recorded at the two highest temperatures tested in their study. In fact, temperature is considered the most important abiotic factor affecting the success of phytophagous insects (*Bale et al., 2002*). Not surprisingly it has also been suggested that invasive species may have a broader and greater physiological tolerance of temperatures than native species sharing the same habitat (for a review see *Zerebecki & Sorte, 2011*). Therefore, knowledge of the effect of challenging or extreme temperatures on invasive insect species could provide key information towards developing spatial and climatic distribution projections for a range of risk assessment applications, and particularly in the context of global warming.

An opportunity to test this is presented here by a comparison of the invasive scarab *Costelytra zealandica* (White) (Scarabaeidae: Melolonthinae) with the closely related non-invasive species *C. brunneum*. These insects are endemic to New Zealand and also occur sympatrically in several places (*Given, 1966*; *Lefort et al., 2012*; *Lefort et al., 2013*). The extended geographical occurrence of *C. zealandica*, and its severe negative impact on agro-ecosystems, suggests that it has reached a high degree of invasiveness within its home range. In fact, based on early observations, this species seems to have become so widespread that it is only absent from a few remote locations of New Zealand (*Given, 1966*) and not reached the status of invader for few others (*East, King & Watson, 1981*). This greatly contrasts with the restricted geographical range of *C. brunneum* which remains confined to a few patchy areas throughout New Zealand, essentially located in the New Zealand Southern Alps (*Lefort, 2013*).

The widespread distribution of *C. zealandica* in New Zealand is likely to be, in part, related to its tolerance of the wide range of soil temperatures within the array of those encountered throughout New Zealand (*Lefort, 2013*), from sun-baked pastural environments to alpine reaches. As recently suggested by *Sinclair, Williams & Terblanche (2012)* for numerous insect species, this might be made possible if *C. zealandica* has a high degree of phenotypic plasticity enabling it to perform under such variable conditions. Certainly the initial spread of *C. zealandica* within its native range would have been a consequence of the widespread cultivation of exotic host plants such as ryegrass and white clover (*Lefort et al., 2014*). However, it would also have possibly required the species to adjust to an expanded range of soil temperatures in a relatively short time frame. Consistent with this is the conclusion of *Stillwell & Fox (2009)*, that the differential responses of the seed beetle *Stator limbatus* (Horn) to temperature, which impacted on

survivorship, body size and fecundity, was due to a high degree of phenotypic plasticity, rather than on genetic adaptation resulting from long-term evolution. That temperature may also have had a direct influence on the differences in the distribution of *C. zealandica* and *C. brunneum* is corroborated by the empirical study of *Zerebecki & Sorte (2011)* on temperature tolerance and stress proteins, which concluded that invasive species tend to live within broader habitat temperature ranges and higher maximum temperatures.

Several enzyme systems have been successfully linked to, or are suspected to play a key role in, animal physiological tolerance to temperature. Lactate dehydrogenase-B (LDH-B), for example, has been linked to thermal tolerance in a killifish species (*Johns & Somero, 2004*; *Dalziel, Rogers & Schulte, 2009*). The adapting kinetic properties of the cytosolic malate dehydrogenase (cMDH) enzyme have been related to warm temperature adaptation in blue mussels (*Fields, Rudomin & Somero, 2006*), and similarly the isocitrate dehydrogenase locus Idh-1 exhibits significant correlations between allele frequencies and temperature in several species (for a review see *Huestis, Oppert & Marshall, 2009*). In invasive species, *Hanski & Saccheri (2006)* have suggested that the metabolic enzyme system phosphoglucose-6-isomerase (PGI) could play a key role in the expansion and delineation of geographical range boundaries of these species. This enzyme system sits at the intersection of the major glycolysis and glycogen biosynthesis metabolic pathways, catalyzing the second step in glycolysis to energy in the form of ATP to the organism (*Riddock, 1993*). Through this unique biochemical situation, covariation patterns between *Pgi* genotypes and individual fitness performance or life-history traits are considered likely to arise (*De Block & Stoks, 2012*). In fact, phenotypic variability of the PGI enzyme system has been correlated many times to insect fitness, and several such studies on the Glanville fritillary butterfly might also be relevant to success of an invasive species. For example, *Haag et al. (2005)* and *Niitepõld et al. (2009)* established that genetic variation in *Pgi* was correlated with flight metabolism, dispersal rate and metapopulation dynamics in this butterfly. Additionally, *Hanski & Saccheri (2006)* showed that the allelic composition of the PGI enzyme system had a significant effect on the growth of local populations. The link between lifespan duration and the PGI genotype showing high dispersal capacity was also demonstrated (*Saastamoinen, Ikonen & Hanski, 2009*). However, of relevance to the present study, PGI genotype has been designated several times as a key enzyme candidate in insect thermal tolerance to extreme temperatures (for a review see *Kallioniemi & Hanski, 2011*). As such, it has been characterised as the best-studied metabolic enzyme in a recent review of variation in thermal performance in insect populations (*Sinclair, Williams & Terblanche, 2012*). Despite this, this enzyme system has never been analysed in a comparative study involving invasive versus non-invasive insect species.

As part of a wider investigation into invasiveness of phytophagous insects, this study aimed to test the hypothesis that *C. zealandica* is more tolerant of a wider range of temperature than the closely related and co-occuring non-pest species *C. bruneum*, thus facilitating its establishment over a wider geographic area, and to investigate whether particular PGI-genotypes are related to individual fitness advantage when exposed to challenging soil temperatures.

## MATERIAL AND METHODS

### Insect sampling and identification

Young, actively feeding, third instar larvae were collected; one population of *C. zealandica* from the South Island of New Zealand (Hororata, 43°32′17″S 171°57′16″E) and from the North Island of New Zealand (Te Awamutu, 38°09′95″S 175°35′07″E), and one population of *C. brunneum* from the South Island of New Zealand (Castle Hill, 43°12′20″S 171°42′16″E). Larval sampling was performed by using a spade to turn over the upper 20 centimeters of soil at randomly chosen locations at each collection site, as described in (*Lefort, 2013*). The three collection sites were respectively labeled A, B and C. Larvae were identified to species based on the methodology described in *Lefort et al. (2012)*, *Lefort et al. (2013)*. Fewer *C. brunneum* were found compared to *C. zealandica*. Prior to experimentation all larvae were tested for amber disease, which commonly occurs in *C. zealandica*, as described in *Jackson, Huger & Glare (1993)* and only healthy larvae were used.

### Survival and growth response to different temperature regimes

*Costelytra* larvae usually live at an average soil depth of 10 cm (*Wright, 1989*). At this depth, and because of the resulting buffer effect, the yearly maximum temperatures rarely reach 20 °C and often remain above 5 °C during the coolest months of the year in New Zealand (NZ Meteorological Service 1980). Because of the univoltine nature of the *Costelytra* species life-cycle (*Atkinson & Slay, 1994*), feeding third instar larvae are rarely exposed to soil temperatures below 10 °C for long periods. Therefore 10 and 20 °C were used as realistically challenging temperatures within the normal soil temperature range for these species, while a 15 °C standard laboratory rearing temperature (*Lefort, 2013*) was used as control.

The larvae of each population ($n = 90$ for each *C. zealandica* population, and $n = 30$ for *C. brunneum* population) were randomly allocated to one of the three temperature treatments at which each larva was reared individually as described in *Lefort et al. (2014)*. All larvae were fed ad libitum with chopped roots of *Trifolium repens* (white clover).

Larval survival and growth measured as weight gain were recorded as measures of fitness, and assessed weekly over a period of six weeks. Larvae were carefully taken out from their containers and handled with soft forceps. They were weighed on a 0.01g readability portable Sartorius TE412 digital scale. Dead larvae were collected every 24 h and individually stored at −80 °C to minimise protein degradation for the electrophoretic study. At the end of the experiment, all the larvae were snap frozen and similarly stored.

### PGI electrophoretic study

The last abdominal segment of each larva was cut into small pieces on a square glass plate over ice and then ground using an autoclaved plastic rod in 100 µl of cooled extraction buffer (Tris–HCl, pH 8.0) until completely homogenized.

Expression of the PGI allozymes was subsequently examined by cellulose acetate electrophoresis according to the manufacturer instructions (Helena Laboratories, Beaumont, US) and following optimization of the method of *Hebert & Beaton*'s (*1993*).

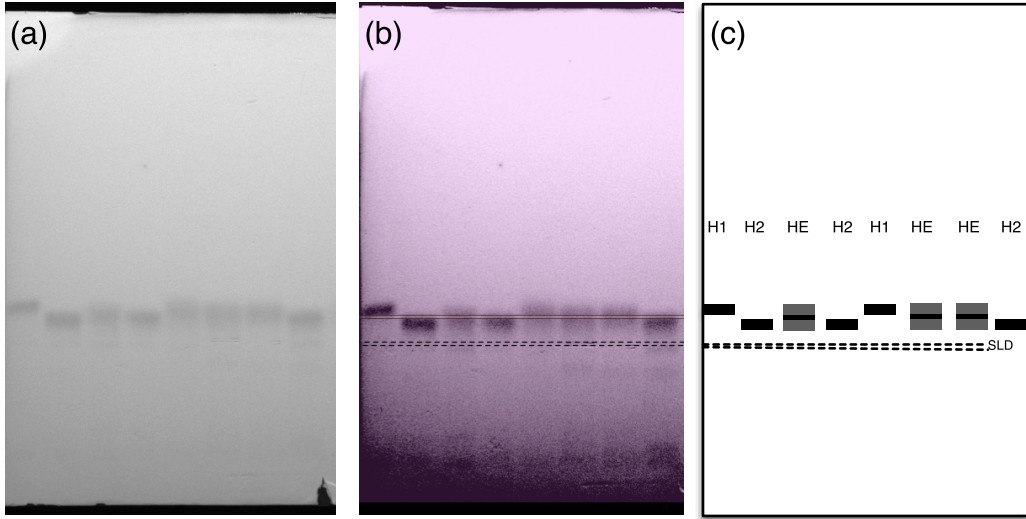

**Figure 1 Example of scoring of a cellulose acetate plate.** Where (A) is an original picture of the plate, (B) is a definition optimisation and alignment of the allozyme profiles via Adobe Photoshop CS5 and OmniGraffle 5 Professional and (C) is a representation of the allozyme profiles. H1, fast allele detected—homozygote 1; H2, slow allele detected—homozygote 2; HE, fast and slow allele detected—heterozygote; SLD, sample loading zone.

The final procedure comprised of 10 µl of each homogenate electrophoresed on cellulose acetate plates (Titan® III 76 mm × 76 mm; Helena Laboratories) in Tris-Glycine electrode buffer pH 8.5 at a constant voltage of 200 V and 2 mA for 15 min. A positive heterozygote control for each *Costelytra* species was run on each plate. Plates were immediately stained with 4 ml of a freshly prepared PGI stain mix (*Hebert & Beaton, 1993*). Staining time was estimated visually and lasted between 1 and 2.5 min. Plates were then soaked for 30 min in water, blotted dry and preserved by incubating at 60 °C for 15 min.

Each plate was subsequently digitised using a UVIDOC HD2 (Uvitec Cambridge, UK) and band scoring performed by optimising the definition and aligning the different allozyme profiles obtained using Adobe Photoshop CS5 and OmniGraffle 5 Professional (Fig. 1). For each population studied, heterozygote and homozygote forms were scored for each population studied; homozygote alleles were assigned as slow or fast based on their relative mobility from the loading zone (1C).

## Data analyses

Statistical analyses to determine the effect of temperature on larval survival were carried out using Fisher's exact tests. Comparisons to determine differences of larval survival rates in relation to populations, allozyme(s) genotype and temperature regimes, were carried out using Fisher's exact tests (Fig. 2).

To detect significant differences in larval growth rate among temperature treatments, growth data were first analysed by analysis of variance (one way ANOVA), followed by Least Significant Difference (LSD) post-hoc analysis after exclusion of larvae that died before the end of the six week period.

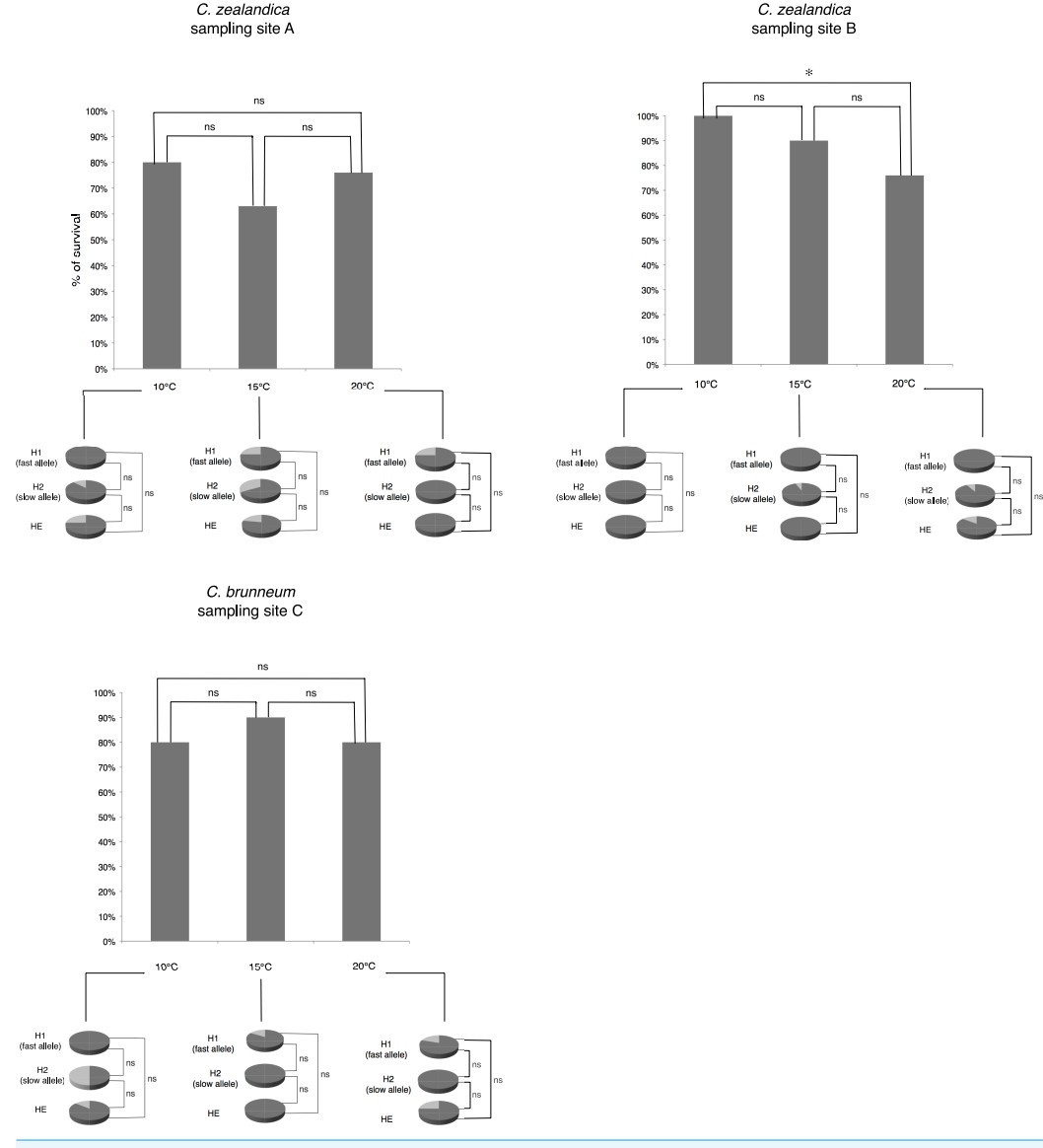

**Figure 2  Percentage of larval survival of *Costelytra zealandica* and *C. brunneum* after 6 weeks of treatment under different temperatures (10, 15 and 20 °C).** Details of larval survival (dark grey)/mortality rates (light grey) observed for each PGI-genotype detected in each population.

For each population, a factorial ANOVA was carried out to test if the interaction term, temperature treatment × PGI genotype, had an effect on larval growth. When a significant temperature treatment × PGI genotype effect was detected, follow-up T-tests were performed to compare the average growth of each PGI genotype under each of the three temperature regimes.

Statistical tests were conducted with R software (*R Development Core Team, 2011*) and GenStat® (GenStat 14; VSN International Ltd, UK).

**Table 1** PGI genotypes detected by cellulose acetate electrophoresis in *Costelytra zealandica* and *C. brunneum* and their effective distribution in each temperature treatment.

| | H1 (homozygote - fast allele) | H2 (homozygote - slow allele) | HE (heterozygote) |
|---|---|---|---|
| *C. zealandica* (sampling site A) | $n = 14$ (respectively $n = 8$, 3 and 3 at 10, 15 and 20 °C) | $n = 16$ (respectively $n = 7$, 2 and 7 at 10, 15 and 20 °C) | $n = 36$ (respectively $n = 9$, 14 and 13 at 10, 15 and 20 °C) |
| *C. zealandica* (sampling site B) | $n = 13$ (respectively $n = 5$, 3 and 5 at 10, 15 and 20 °C) | $n = 43$ (respectively $n = 17$, 15 and 11 at 10, 15 and 20 °C) | $n = 23$ (respectively $n = 8$, 9 and 6 at 10, 15 and 20 °C) |
| *C. brunneum* (sampling site C) | $n = 10$ (respectively $n = 1$, 5 and 4 at 10, 15 and 20 °C) | $n = 4$ (respectively $n = 1$, 2 and 1 at 10, 15 and 20 °C) | $n = 11$ (respectively $n = 6$, 2 and 3 at 10, 15 and 20 °C) |

**Table 2** Factors affecting *Costelytra* spp. larval growth rate and their interactions.

| | *C. zealandica* (sampling site A) | | | | *C. zealandica* (sampling site B) | | | | *C. brunneum* (sampling site C) | | | |
|---|---|---|---|---|---|---|---|---|---|---|---|---|
| Variable | F | df | P | Significance | F | df | P | Significance | F | df | P | Significance |
| Temperature | 5.5472 | 1 | **0.0218** | ☞ | 1.6051 | 1 | 0.2205 | ns | 2.4125 | 1 | 0.1247 | ns |
| PGI genotype | 0.2132 | 2 | 0.8086 | ns | 1.0164 | 2 | 0.3807 | ns | 0.8143 | 2 | 0.4469 | ns |
| Temperature × PGI genotype | 7.6109 | 2 | **0.0011** | ☞ ☞ | 0.1573 | 2 | 0.8555 | ns | 1.8284 | 2 | 0.1679 | ns |

## RESULTS

Few larvae died during the course of the experiment and were not used in the analysis. All surviving larvae produced scorable electrophoretic patterns. Sample sizes are summarized in Table 1. The electrophoretic study revealed the existence of only one PGI-locus in both *Costelytra* species (Fig. 1). The genotypes along with their distribution in each temperature treatment are summarized in Table 1.

There were no significant differences in larval survival under the different temperature treatments for populations A and C. For population B (*C. zealandica* collected from the North Island of New Zealand), survival was significantly higher at 10 °C (100% survival) than it was at 20 °C (73% survival) (Fisher's exact test, $P = 0.0046$) (Fig. 2). However, the weight gain of that population was not significantly different for any temperature treatment (Fig. 3).

In contrast, the weight gain of South Island *C. zealandica* (site A) significantly increased under the highest temperature of 20 °C, compared to the lowest temperature (Fig. 3), while *C. brunneum* gained, on average, significantly more weight at 15 °C compared to the more challenging extremes (Fig. 3).

A significant interaction between temperature treatment and PGI genotype on larval growth was only detected for *C. zealandica* collected from the South Island (sample site A) (Table 2).

In this population, at 10 °C, homozygotes H1 (fast allele) gained significantly more weight compared to homozygotes H2 (slow allele) (Fig. 4). On the other hand, under the

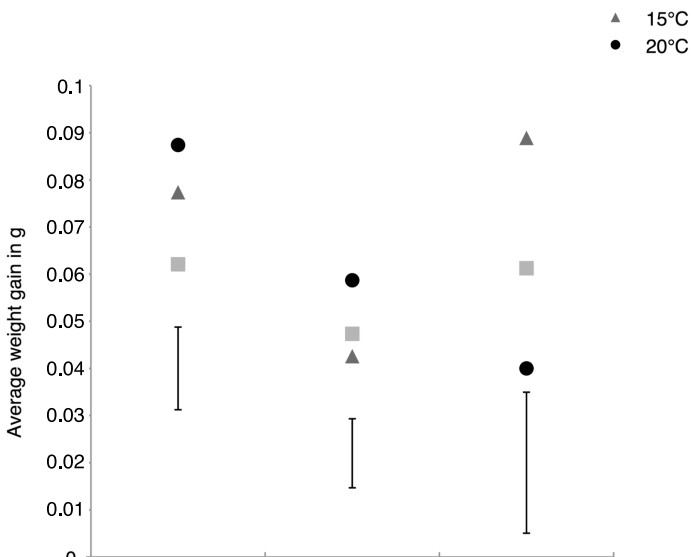

**Figure 3** **Average weight gains of the surviving larvae of two populations of *Costelytra zealandica* and one population of *C. brunneum* after 6 weeks of treatment under different temperatures.** Vertical bars represent the 5% Least Significant Difference (LSD); LSDs = 0.0175, 0.0146 and 0.0299 for the populations from sample sites A, B and C respectively.

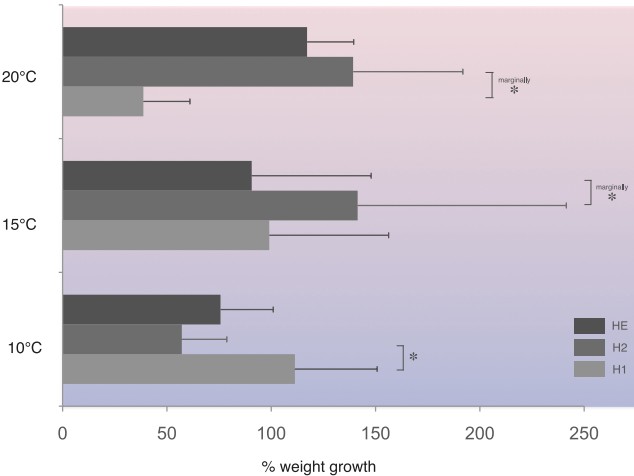

**Figure 4** **Average percentage weight gain in *Costelytra zealandica* larvae collected from the New Zealand South Island (sample site A).** Weight gain is presented for each PGI genotype (H1, fast allele—homozygote 1; H2, slow allele—homozygote 2; HE, both alleles–heterozygote), after six weeks at 10, 15 and 20 °C. Horizontal bars represent SE. All pairwise comparisons were performed using Student's T-test. Asterixes indicate significant (*) (Student's T-test, $P < 0.05$), or marginally at 1% level of significance (marginaly *) differences (Student's T-test, $P < 0.10$).

higher temperature regime tested, homozygotes H2 gained significantly more weight at the 1% level of significance compared with homozygotes, H1, at 20 °C and heterozygotes, HE, at 15 °C (Fig. 4).

## DISCUSSION

The main objective of this study was to investigate the suggestion proposed by *Hanski & Saccheri (2006)* that the *Pgi* gene may be strongly implicated in '*the expanding front of invasive species*'. Towards this the present study aimed at testing whether a relationship between the PGI-genotypes of *Costelytra* species and their fitness response under different temperature regimes exists. However, the results here did not consistently indicate such correlation across all the populations tested, although significant effects of temperature on the larval growth and/or survival of the two species were detected.

Consistent with the latter, *Lefort (2013)* had previously shown that *C. zealandica* had an improved survival rate at lower rather than at higher temperatures. In the present study this was only apparent for the North Island population, having better survival at 10 °C compared with the highest temperature of 20 °C. This discrepancy, of no effect here for the South Island population, could be due to the relatively short length of this study compared to the 17 weeks of *Lefort (2013)*. Additionally, the North Island larvae used here could have been more sensitive to the experimental conditions, because they were subject to a higher degree of disturbance and stress associated with longer transport from the sample site to the laboratory.

In a similar way to larval survival, the temperature effect on larval growth was significantly different between the two populations of *C. zealandica*. Weight gain of the larvae collected from the South Island was significantly depressed by low temperature (i.e., 10 °C), whereas this effect was not detected in the larvae collected from the North Island. Some degree of genetic divergence between the two populations studied might exist and explain this disparity. For example, the cosmopolitan *Drosophila melanogaster* Meigen exhibits complex patterns of genetic variation between populations that have allowed it to successfully establish worldwide under extremely diverse thermal environments (*Hoffmann, Sørensen & Loeschcke, 2003*; *Morgan & Mackay, 2006*). There is in fact some evidence of genetic divergence within *C. zealandica* based on the ITS1 rDNA sequences reported by *Richards Glare & Hall (1997)*, particularly between North Island and South Island populations. Furthermore, adult specimens of *C. zealandica* have been reported to be larger in the North Island (T Glare, pers. comm., 2009, Bio-Protection Research Centre NZ). In addition, *Lefort et al. (2014)* have demonstrated that host-race formation in this species might have been instrumental to its invasion success. This may have contributed to the establishment of even further genetic divergence between *C. zealandica* populations throughout New Zealand since that reported by *Richards Glare & Hall (1997)*, although in the present study both populations were collected from exotic pastures mostly composed of the same host plants.

The non-pest species *C. brunneum*, in contrast to *C. zealandica*, showed significantly impaired, but not lethally detrimental, larval growth under the most challenging temper-

ature regimes, particularly at 20 °C. Even though very little is known about the actual distribution of *C. brunneum*, this species seems to prefer mid to high altitudinal ranges (*Hoy & Given, 1952*; *Given, 1966*). In those regions soil temperatures are likely be similar to the averages recorded in the coldest southern locations of New Zealand, e.g., Invermay's yearly average soil temperature ranged between 15 and 2.9 °C (NZ Meteorological Service 1980). These observations corroborate the fact that this non-invasive species is less tolerant than *C. zealandica* to challenging temperatures, particularly higher temperatures, which would be consistent with its failure to extend its geographical range as has its invasive congener.

Because of its potential impact on the functional properties of metabolic enzymes (*Kallioniemi & Hanski, 2011*), temperature is often characterised as a key environmental factor affecting the growth and survival of poikilothermic organisms (*Kallioniemi & Hanski, 2011*; *Sinclair, Williams & Terblanche, 2012*). Hence, the interest in the expression of various forms of these enzymes, their allozymes or isoenzymes, has considerably increased over recent decades (*Karl, Hoffmann & Fischer, 2010*). Amongst these, PGI has been described as a highly polymorphic enzyme system in numerous taxa (*Kallioniemi & Hanski, 2011*). For instance, seven alleles were detected for the *pgi* gene in *Melitaea cinxia* (Linnaeus), the Glanville Fritillary butterfly, and many coleopteran species possess over three alleles for this gene (e.g., *Nahrung & Allen, 2003*; *Dahlhoff & Rank, 2007*). Such polymorphism provides potential for species to vary in their ecological response, including their thermal tolerance (*Kallioniemi & Hanski, 2011*). In the present study, the electrophoretic profiles revealed the expression of two alleles at only one PGI-locus for these *Costelytra* species (Fig. 1). However, as the resolution using cellulose acetate technology is not high, there is the possibility that detection of additional alleles was missed because of poor migration and separation of the various allozymes on the gel, or that allozymes of different loci have a similar or highly similar net charge rendering them indistinguishable under these electrophoretic conditions. Alternative higher resolution methods such as mass spectrometry, or genetic and genomic methods may be needed to confirm this enzyme system is not as polymorphic as it appears in other insect species.

It was expected that if PGI-associated metabolic pathways provide a greater ability to adapt to a wider range of temperatures, heterozygote individuals for this enzyme system would have displayed better weight gain and survival rates under challenging temperatures than homozygotes. Such results would be consistent with Watt's studies (*1977*; *2003*) of the PGI enzyme system in *Colias* butterflies, which reported heterozygotic advantage with respect to several life-history and fitness traits in this species. The results of the present study showed that homozygote genotypes H1 gained more weight under the lowest temperature tested, and also suggested higher fitness in homozygotes H2 under higher temperatures in the *C. zealandica* collected from the New Zealand South Island site. However, despite these counterintuitive results, it is important to bear in mind that extensive studies on PGI expression in several butterfly species have demonstrated that enhanced individual performances are correlated with different allelic compositions for various life-history traits. For instance, *Karl, Schmitt & Fischer (2008)* and *Karl, Schmitt & Fischer (2009)* demonstrated that there was enhanced larval and pupal growth and

development in heterozygote genotypes PGI 2-3 in the butterfly *Lycaena tityrus*, whereas cold stress resistance in the same species was associated with a different PGI genotype. Hence, heterozygote advantage may not systematically apply to this enzyme system. Furthermore, the two above studies by *Karl, Schmitt & Fischer (2008)* and *Karl, Schmitt & Fischer (2009)*, could suggest that allelic variability in the PGI enzyme system is expected to be higher in invasive species compared with non-invasive ones, since successful invaders often display high individual performance in many different fitness and life history traits. Therefore, if the low allelic variability observed in *C. zealandica* in this study is the result of a misinterpretation of the electrophoresis profiles, this could in part explain why no consistent relationship between individual larval fitness responses and PGI genotypes between the two populations of the invasive species were detected.

Additionally, the fact that a significant relationship was detected between the selected life-history traits and the PGI genotypes in only one population of the species studied may be due to the experimental design being driven by the need to perform the temperature experiments prior to sacrificing the larvae for the electrophoretic study. This compromised design, using individual insects of unknown allelic composition, has resulted in small and unbalanced sample sizes as shown in Table 1.

In conclusion, the present study has been unable to support the hypothesis that the *Costelytra* spp. response to challenging temperatures was indeed related to the *pgi* gene and more precisely with PGI allozyme forms expressed by this gene. Small and unbalanced sample size, with respect to allele types, along with the low allelic variability in *Costelytra* species and the resulting difficulties to interpret the electrophoretic profiles could explain why no relationship between this gene and thermal tolerance in the studied species was found. Indeed, other studies have established a link between thermal tolerance and the *pgi* gene in various species, including cnidarians (*Zamer & Hoffmann, 1989*), beetles (*Dahlhoff & Rank, 2000*) and moth and butterfly species (*Karl, Schmitt & Fischer, 2009*; *He, 2010*). It is also important to note that the present study investigates the underground life-stage of the targeted species. The thermal dynamics of these insects are therefore somehow different from those of aboveground insects such as adult willow leaf beetles or butterflies that experience a much wider range of temperatures on a daily basis than underground life-stages, and which have until now served as model systems for studying the effects of the PGI system on life history traits. In addition, this study was centered on insect survival and growth, while other life history and fitness traits, relevant to insect invasion success such as for instance dispersal capacity and mating/offspring successes, might also be affected by the PGI system. For these reasons, we believe that it is still possible that differential expression of this gene could be involved in the invasion success of some insects, even maybe so in the adult stage of *C. zealandica*, allowing them to extend their range over wider geographical areas compared to other species.

Several studies have successfully linked various forms of PGI allozymes with the expression of heat shock proteins (Hsps), which play important roles in thermal tolerance by reducing stress-induced protein aggregation (*Dahlhoff & Rank, 2000*; *Dahlhoff & Rank, 2007*; *McMillan et al., 2005*). Additional investigations on Hsps expression in *Costelytra*

spp., rather than on the PGI enzyme system itself, could help to establish whether the tolerance to challenging soil temperatures observed in the invasive species *C. zealandica* somehow relates to the PGI enzyme system. The sympatric nature of the non-invasive and invasive species studied here provided a valuable opportunity to investigate PGI as a marker of invasiveness. There are many other such species pairs such as the queensland and lesser queensland fruit flies that would serve the same purpose. Therefore we strongly encourage researchers to replicate the experiments described in this paper using such invasive/non-invasive species pairs and to confirm whether or not a relationship exists between the PGI enzyme system and insect fitness response to temperature and with the potential to be used as a measure of environmental tolerance in invasive species.

## ACKNOWLEDGEMENTS

We would like to thank Richard Townsend and St Andrew's College of Christchurch for granting access to the collection sites, and Derrick Wilson for his help with the collection of the larvae from the North Island. Finally we would like to thank the reviewer for his constructive comments and advice on the initial manuscript.

### Funding

Financial support was provided by the Miss EL Hellaby Indigenous Grasslands Research Trust, Better Border Biosecurity and the Bio-Protection Research Centre. The funders had no role in study design, data collection and analysis, decision to publish, or preparation of the manuscript.

### Grant Disclosures

The following grant information was disclosed by the authors:
Miss EL Hellaby Indigenous Grasslands Research Trust.
Better Border Biosecurity.
Bio-Protection Research Centre.

### Competing Interests

All the authors are employees of the Bio-Protection Research Centre.

### Author Contributions

- Marie-Caroline Lefort conceived and designed the experiments, performed the experiments, analyzed the data, wrote the paper, prepared figures and/or tables, reviewed drafts of the paper.
- Samuel Brown performed the experiments.
- Stéphane Boyer, Susan Worner and Karen Armstrong reviewed drafts of the paper.

### Supplemental Information

Supplemental information for this article can be found online at http://dx.doi.org/10.7717/peerj.676#supplemental-information.

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
