# Peer review of "The PGI enzyme system and fitness response to temperature as a measure of environmental tolerance in an invasive species"

_PeerJ, doi:10.7717/peerj.676_

## Round 0.1 · original submission · Major Revisions

The work is well written and the hypothesis raised is very interesting, but authors must pay attention to the points remarked by the reviewer who has made a detailed specification of the necessary changes.
Pay special attention to:
1) Include more detail about the measurement of variables such larval weight
2) Reorder the description of the statistical analysis section in one section
3) Analyze the usefulness of Figure 2 and its change by a Table
4) Add some of the statistical analysis suggested by the reviewer, which will clarify the results
5) Discuss the Validity of the findings in the terms pointed out by the reviewer
6) Review the References
Also minor editorial changes are included by the reviewer.

·

Basic reporting

This is an interesting manuscript which focusses on the effects of temperature and phosphoglucose isomerase (allozyme) genotype on larval performance in two scarab beetles. The study is placed in the context of invasive species and one of the hypotheses tested is whether the invasive beetle species has broader thermal tolerance than the non-invasive one.

The manuscript is generally well written, and the experiment is clearly described. I didn’t find how the larvae were weighed, though, so the authors should add that to the Material and Methods. It would also be helpful to describe all the statistical methods in one section, instead of in the context of each experiment. I was struggling a little bit with some of the results as it wasn’t always clear what was included in the analyses, and I have some questions regarding those (below). In general, the authors don’t overinterpret their results, which I like.

What is particularly interesting here is that the larvae live underground and, as the authors state, they are therefore not exposed to extreme temperatures. The thermal dynamics of these beetle larvae are thus very different from adult willow leaf beetles or butterflies, which have served as model systems for studying the effects of phosphoglucose isomerase. Temperate butterflies experience a much wider range of body temperatures on a daily basis, and can up or down-regulate their body temperature by approximately 10 C by simply adjusting their wing and body orientation in relation to the sun. In this light, the lack of a strong PGI effect on larval growth rate in this study might not be that surprising, as the temperatures experienced during the larval stage might not have a huge effect on overall fitness. In future studies it would be interesting to see if PGI would have effects on adult performance and fitness in this study system.

The study did find effects of temperature on larval growth rate in one C. zealandica population and in the non-invasive C. brunneum. In its current state the manuscript does not include a statistical test examining if this effect differs between the three PGI genotypes. Adding this test would be critical for being able to answer the question of the importance of PGI in influencing insect life history and fitness.

I found the title quite long and complex. Any chance the title could be revised?

I had a few questions and comments on the Results section:

Results, page 7, PGI electrophoretic study: Did you test if the allele frequencies deviate from the Hardy-Weinberg equilibrium? Looks like there might be something going on in the North Island population. The sample size reported here is smaller than the initial sample size. Were you able to genotype dead larvae, like you suggest in the Material and Methods?

Results, page 7, line 195 onwards: You used Fischer’s exact test for analysing larval mortalities. How exactly was this test carried out? Did you pool all the genotypes for each temperature treatment? In Fig. 2 the genotypes are displayed separately but I guess this is only for illustrative purposes? I actually find Fig. 2 quite difficult to read, since the bars are based on absolute numbers, not percentages. For example, a quick glance at C. zealandica from site A, temperature 15 C shows that PGI heterozygotes had both the highest survival and highest mortality values compared to the homozygotes. This is of course because among the survivors there were 14 heterozygotes and only 3 and 2 fast and slow allele homozygotes, respectively. What really would be interesting would be seeing the relative survival figures for each treatment and genotype (coupled with sample size). I would suggest you ditch Fig. 2 and make a table instead.

Results, page 8, line 206 onwards: I understand you used ANOVA for detecting significant differences in larval growth rate among the temperature treatments, and if significant differences were found, LSD to find out which groups were significantly different. Please provide the statistics in the text, in a table or incorporated in Fig. 3. Speaking of Fig. 3, the error bars are somewhat unorthodox. It might be more useful to add separate error bars to each mean value symbol. Consider revising the entire figure.

However, isn’t this test essentially the same as in the next section (and Table 2), but without PGI genotype? For clarity, I would suggest rearranging the results section so that larval survival and larval growth would be separated. In other words, move the part describing larval growth from the second section to the third section.

Results, page 9, line 219 onwards: You pooled the homozygotes in this analysis. Could you clarify why you did this? I believe the reason was gaining more statistical power, but you should justify this in the text. The slow and fast homozygotes could have very different phenotypic properties.

Calling the (pooled) PGI genotype a co-variate is a bit confusing, as with covariates people most often refer to continuous variables in an ANCOVA. In table 2, it would anyway be clearer to use the term ‘PGI genotype’ instead of ‘Co-variate’. That way the casual reader doesn’t have to read the legend to see what the variable is.

Now to an important point: Did you test if the interaction between temperature treatment and PGI had an effect on larval growth rate? This should be the most interesting test when testing your hypothesis. If you did include the interaction in your initial models, but it turned out to be non-significant, this should be mentioned. If space permits, it would be nice to see a figure with these results. The trouble here is that with the low and unbalanced sample size, this most critical test might not be that powerful.

Experimental design

Unfortunately the study suffers from two problems in the experimental design: low number of sampling locations and low number of individuals sampled. The first one isn’t as serious but affects the generality of the results. As all C. zealandica samples were collected from only one population on each island, it is impossible to tell whether differences between the two populations reflect larger geographical patterns or just random local variation. This isn’t necessarily a huge problem, since the focal question of the study is the interplay between temperature and PGI, not assessing geographic variation. However, one has to be a little bit cautious when interpreting the results, as in the Glanville fritillary butterfly study system (as the authors mention in the paper) PGI frequencies in local populations change with time since colonisation and PGI interacts with habitat patch size in affecting population growth rate. This means that the selection pressure on specific traits varies in time and space, and general conclusions are difficult based on just one sampled local population.

The problem with low sample size is more serious, as it affects the validity of key results of the study. One of the main purposes of the study is to assess the role of PGI on larval growth rate and mortality. In other study systems, individuals with different PGI genotypes have been shown to differ in thermal sensitivity, which leads to fitness consequences. Considering temperature and PGI together is therefore critical, but the low sample size of this study makes it difficult. A total of 90 sampled larvae from each C. zealandica would under ideal conditions leave 30 individuals to each temperature treatment. If the distribution of PGI genotypes would be ideal, there should be 15 heterozygotes and 7.5 individuals representing each homozygous genotype. Seven or eight individuals in each treatment-genotype combination might be just enough, but in practice the numbers are significantly lower in many groups. As suggested above, I would nevertheless go ahead and analyse the effect of the PGI by temperature interaction and also plot the data. If the results were consistent there might be clear patterns emerging even with the low sample size.

Validity of the findings

As discussed above, at this stage we don’t know if there would be an interaction between PGI genotype and temperature in affecting larval survival or growth, so that question is still up in the air. This analysis might be problematic because of the low sample size, but I hope the authors can address this nevertheless.

One should also bear in mind that larval survival and growth is just one component of the life history of an invasive insect. The authors state in the abstract: “No relationship between PGI phenotypes and fitness was detected, suggesting that the PGI may not be contributing to the invasion success and pest status of C. zealandica.” However, it is still entirely possible that there could be significant effects of PGI on adult performance. Without knowledge of PGI effects on adult dispersal capacity and egg laying success, it might be too early to say that PGI does not affect the “invasion success and pest status of C. zealandica.” At this stage we simply do not know. The authors should therefore make sure to stress that these findings only apply to the larval stage.

Additional comments

Minor points and specific comments

Title: scarabs -> scarabs’

Abstract, 1st line: species -> species’

Abstract, line 11: scarab -> Perhaps you could use “scarabaeid” beetle here for additional clarity.

Abstract, last line: the PGI -> PGI

Introduction, page 2, line 16: a species -> a species’

Introduction, page 3, line 85: “insect fitness performance” The word “performance” seems redundant here, unless you want to say “performance and fitness” (sensu Feder and Watt 1992)

Introduction, page 3, line 87: Technically speaking the seminal Haag et al. 2005 paper didn’t address all these questions. You could consider adding Niitepold et al. 2009 Ecology here.

Introduction, page 4, line 91: PGI phenotype -> PGI genotype

Material and Methods, page 4, line 114: “Fewer C. brunneum were able to be found compared to C. zealandica.” -> “Fewer C. brunneum were found…” or try using active voice instead of passive.

Material and Methods, page 6, line 165: were assignment -> were assigned

Material and Methods, page 6, lines 175, 177 and 180: phenotype -> genotype

Results, page 7, line 189: phenotypes -> genotypes

Results, page 7, 196: There was -> There were

Figure 2 legend: The legend says that larval survival is indicated by dark grey bars, but at least this version of the figure has reddish shades for survival.

Figure 3: The y axis has decimal commas instead of points.

Results, page 9, line 220: PGI phenotypes -> PGI genotypes


Discussion, page 11, line 280-283: “Because of the potential impact of temperature on the functional properties of metabolic enzymes (Kallioniemi & Hanski 2012), it is often characterised as a key environmental factor affecting the growth and survival of poikilothermic organisms (Kallioniemi & Hanski 2012, Sinclair et al. 2012).” Please revise this sentence, e.g.: “Because of its potential impact on the functional properties of metabolic enzymes (Kallioniemi & Hanski 2012),temperature is often characterized…”

Discussion, page 11, line 287: Granville -> Glanville

Discussion, page 11, line 297-299: “Alternative higher resolution methods such as mass spectrometry may be needed to confirm this enzyme system is not as polymorphic as it appears in other insect species.”
Obviously genetic and genomic methods are even more powerful in assessing the underlying genetic variation, or differences at the expression level. As the authors note, allozyme studies have unfortunately serious limitations, especially since they can’t prove that there aren’t cryptic genotypes.

Discussion, page 12, lines 314-320: Good point, it is unfortunate that this study design can lead to limited genetic representation and unbalanced distribution. On the other hand, this way there is no way to accidentally or intentionally affect the results during the experiment, so performing the experiment this way can also be seen as a positive thing…

Acknowledgements: No acknowledgements?

References

Check that all scientific names are in italics.

Watt WB (2003) Mechanistic stidies of butterfly adaptations. Ecology and evolution taking flight: butterflies as model systems (Eds CL Boggs, WB Watt & Ehrlich PR) University of Chicago Press, New York.
-stidies -> studies

Watt WB (1977) Adaptation at species loci. I. Natural selection on phosphoglucose isomerase of colias butterflies: biochemical and population aspects. Genetics 87:177–194.
-species ->specific
-colias -> Colias

---

## Round 0.2 · accepted · Accept

You have included most suggestions and clarified the manuscript. I accept the new figure and agree that its much better than the previous one. Your paper now is ready to be published.